# Hollow Light Guide Module Involving Mini Light-Emitting Diodes for Asymmetric Luminous Planar Illuminators

**Zhi Ting Ye [1]**, **Chin Lung Chen [2]**, **Lung-Chien Chen [3],***, **Ching Ho Tien [3]**, **Hong Thai Nguyen [4]** and **Hsiang-Chen Wang [4],***

[1] Department of Electro-Optical Engineering, National United University, 2, Lienda, Miaoli 26063, Taiwan
[2] Materials and Chemical Engineering, National United University, 2, Lienda, Miaoli 26063, Taiwan
[3] Department of Electro-Optical Engineering, National Taipei University of Technology, No. 1, Sec. 3, Chung-Hsiao E. Rd., Taipei 10608, Taiwan
[4] Department of Mechanical Engineering and Advanced Institute of Manufacturing with High-Tech Innovations, National Chung Cheng University, 168, University Rd., Min-Hsiung, Chia-Yi 62102, Taiwan
* Correspondence: ocean@ntut.edu.tw (L.-C.C.); hcwang@ccu.edu.tw (H.-C.W.)

**Abstract:** Light-emitting diodes (LEDs) have numerous advantages. However, LEDs only offer a point light source. Therefore, transforming LEDs into planar light sources is a new objective in general lighting applications. Solid light guides have strong uniformity but are marred by their material absorption characteristics and weight. Hollow light guides constitute a solution to the weight problem but exhibit poor uniformity and necessitate sacrificing efficiency to enhance uniformity. To resolve the uniformity, weight, and efficiency problems simultaneously, we propose a hollow light guide architecture involving mini-LEDs with asymmetric luminous intensity. To develop this guide module, we first optimized the aspect ratio of the cavity and then modulated the light path by using varied angles of the reflection surface on the end wall of the module. We then designed a beveled reflection surface near the mini-LEDs to further enhance uniformity. An archetype of the proposed architecture for planar light source modules had a width and depth of 51.5 and 9.95 mm, respectively. Experimental results revealed a total efficiency of 83.9% and uniformity of 92.3%. The module weight was determined to be 215 g, which was 40% lighter than that of similarly sized solid light guide modules.

**Keywords:** planar illuminators; diffuse reflection cavity; asymmetric; mini-LED

## 1. Introduction

After several decades of light-emitting diodes (LEDs), emergent, LED lighting technology has undergone several significant developments and has gradually become the typical light source for backlighting liquid crystal displays and lighting devices. This revolution has marked a milestone breakthrough in LED technology and led to LEDs becoming an irreplaceable light source. Using LEDs rather than fluorescent lamps as light sources offers several advantages due to the higher color performance, energy efficiency, and eco-friendly characteristics of LEDs [1–3]. Studies have been conducted to test designs for extracting light from backlight modules [4–6]. Recently, planar panel light sources, a new lighting product, have been used in lighting applications. Planar illuminators are usually designed using the concept of side-light backlighting. Such illuminators include LEDs, light guide plates (LGPs), brightness enhancement films (BEFs), and reflection films [7]. Direct-type backlighting modules are another category of backlighting systems, and such modules are relatively thick. Therefore, several studies have been conducted on hollow light guides. The primary problem

encountered in hollow light guides is uniformity. Researchers have attempted to resolve this problem by using a second lens to modulate the luminous intensity of LEDs [8], using multicolored LEDs with micro-lens arrays [9–13], using LEDs with the anodic aluminum oxide structure [14], and using multiple LEDs with various angles of emission toward a reflector [15]. Moreover, researchers have attempted to enhance the uniformity of diffusion plates to avoid partial reflection and transmission of light [16–18]. To modulate the light distribution profile of LEDs to enhance uniformity, a study proposed the use of a second lens with a parabolic surface and different focal lengths [19]. The estimated overall efficiency achieved through the aforementioned designs is 80% [20]. In hollow light guides, reducing the thickness is difficult because of the absence of a solid light guide. Uniformity and efficiency can be enhanced by designing a diffuse reflection cavity that provides multiple reflection rates [21]. Accordingly, this study proposes a hollow light guide architecture involving mini-LEDs with an asymmetric luminous intensity distribution, diffuse reflection cavity, and beveled reflection surface to enhance efficiency, enhance uniformity, and reduce weight.

## 2. Hollow Light Guide Architecture Involving Mini-LEDs with Asymmetric Luminous Intensity

LEDs are point light sources, so the traditional design must be converted to a surface source by using an LGP. Traditional architectures typically include light guides, reflector films, dot patterns, and optical films [22–25]. A light guide plate is mainly made from solid materials such as polycarbonate (PC) and polymethyl methacrylate (PMMA). Using a light guide plate has several disadvantages due to its bulky characteristics in size and weight. Therefore, we propose using varied angles of the reflection surface (VARS) on the end wall in the diffuse reflection cavity. The proposed hollow light guide structure does not have any of these components or optical films. Instead, it uses a brightness enhancement film and diffuser plates. Figure 1 presents a comparison of a solid- and hollow-light guide for a planar light module. The hollow light guide was simulated and optimized to improve the aspect ratio (ASR) of the hollow body and adjust the VARS to a uniform planar surface source.

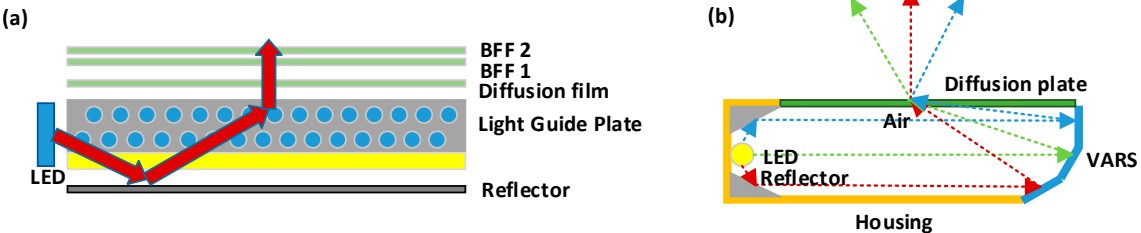

**Figure 1.** (**a**) Solid light guide module for backlighting applications and (**b**) proposed hollow light guide module for backlighting applications.

### 2.1. Simulation and Design of Proposed Hollow Light Guide Involving Mini-LEDs with Asymmetric Luminous Intensity Distribution

A simulation was conducted to demonstrate the effectiveness of the proposed hollow light guide module. Specifically, a ray tracing simulation was performed using TracePro (LAMBDA Corp., Virginia, MA, USA). Figure 2 shows a light source module that contained the proposed hollow light guide and had widths of 51.5 ($W_1$) and 42.8 mm ($W_0$), length of 100 mm ($L_0$), and height of 9.95 mm ($H_0$). The diffuse reflection cavity was composed of aluminum 6061 and was fabricated through extrusion. The mini-LEDs with asymmetric luminous intensity distribution were packaged in a flip chip package [26]. Table 1 lists the surface and material parameters of all components of the light source module used in the simulation.

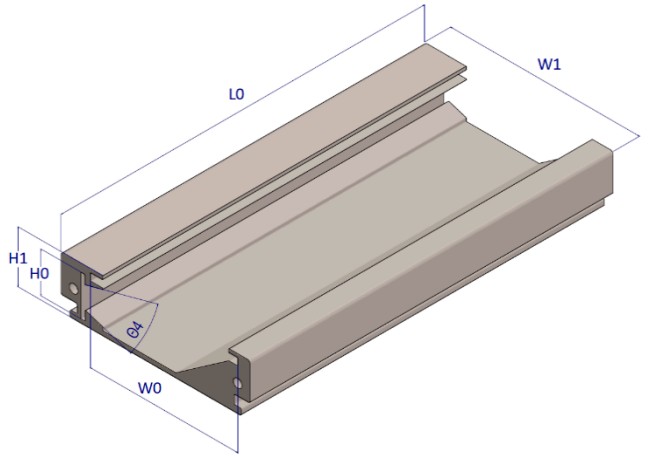

**Figure 2.** Dimensions of light source module.

**Table 1.** Simulation parameter settings.

| Components | Specifications | |
|---|---|---|
| | **Surface Characteristics** | **Material** |
| Diffusion Reflection Cavity | Diffusion white reflection: R = 94% | Aluminum 6061 |
| Varied Angle of the Reflection Surface | Diffusion white reflection: R = 94%<br>BRDF: 0.94<br>BRDF g = 0<br>BRDF B = 0.1<br>BRDF A = 0.329132 | Aluminum 6061 |
| Diffusion plate | BRDF: 0.66<br>BRDF g = 0<br>BRDF B = 0.1<br>BRDF A = 0.231<br>Thickness (mm): 1.5<br>Transmittance ratio: 0.33<br>Diffusion type: Lambertian<br>Manufacturer: Entire Technology<br>Company Limited.<br>Model: EML-R35A | polycarbonate (PC) |
| Metal Core PCB (printed circuit board) | Diffusion white reflection: R = 90% | Aluminum |
| Brightness Enhancement Film | Index = 1.59<br>Surface: polished<br>Prism Angle (degrees): 90<br>Prism Pitch (μm): 50<br>Caliper (μm): 152<br>Thickness (μm): 155<br>Model: 3M AEF-155<br>Manufacturer: 3M$^{TM}$ | polyethylene terephthalate (PET) |

*2.2. Simulation and Design of Optical Cavity of Hollow Light Guide*

The efficiency and uniformity of a light module depends on the aspect ratio of the optical cavity, which can be expressed as $W_0/H_0$, where $W_0$ represents the optical cavity width and $H_0$ represents the optical cavity height. Height is the key factor related to uniformity. Accordingly, we set a constant width but varied the height to evaluate the best aspect ratio of the optical cavity in our simulation. Figure 3 illustrates the hollow light guide module for a planar light source containing mini-LEDs with asymmetric luminous intensity distribution, a brightness enhancement film, and a diffusion plate. Table 2 presents the efficiency and uniformity simulated for different aspect ratio values.

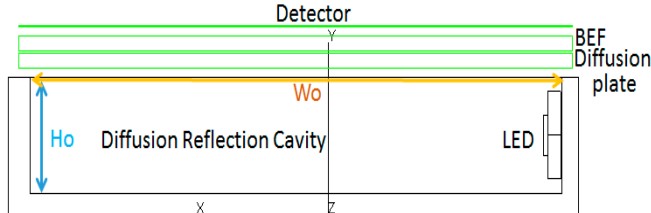

**Figure 3.** Hollow light guide module for planar light source, where BEF is brightness enhancement film.

**Table 2.** Efficiency and uniformity of a light module at various aspect ratio values.

| $W_0$ | $H_0$ | Aspect Ratio ($W_0/H_0$) | Uniformity (%) | Efficiency (%) |
|-------|-------|--------------------------|----------------|----------------|
| 42.8 | 12.5 | 3.4 | 87.1 | 79 |
| 42.8 | 11.5 | 3.7 | 88.2 | 80 |
| 42.8 | 10.5 | 4.1 | 89.4 | 80.2 |
| 42.8 | 9.95 | 4.3 | 90 | 80.9 |
| 42.8 | 9.5 | 4.5 | 88.6 | 81.6 |
| 42.8 | 8.5 | 5.0 | 82.8 | 84.6 |
| 42.8 | 7.5 | 5.7 | 75.4 | 84.8 |
| 42.8 | 6.5 | 6.6 | 61.4 | 85 |
| 42.8 | 5.5 | 7.8 | 50.52 | 85.4 |
| 42.8 | 4.5 | 9.5 | 46.7 | 85.8 |

In this paper, the hollow cavity was used to replace the traditional conductive medium with solid light guide plate. First, we optimized the values of the primary hollow light guide, $H_0$ and $W_0$, where we defined the aspect ratio value as $W_0/H_0$ and the evaluation functions were efficiency and uniformity. The uniformity is expressed in Equation (1) [27], where "minimum" and "maximum" denote the degree of illumination (*E)* that can be measured from a detection surface.

$$\text{Uniformity} = 100\% \frac{\text{minimum } E(lux)}{\text{maximum } E(lux)} \tag{1}$$

Figure 4 shows the relationship between efficiency and uniformity at various aspect ratio values. At an aspect ratio value of 4.3, the observed uniformity was superior to the efficiency. Nevertheless, the efficiency was still reasonably favorable. Figure 5 presents the illumination map observed by the detector at an aspect ratio of 4.3. It can be seen from Table 2 and Figure 5 that the efficiency was 80.9% and the uniformity was 90%. The efficiency was improved when the hollow body was a rectangular cavity, however, the light was still ejected inside causing loss of efficiency. For further efficiency enhancement, the end wall of the light guide was tilted at an angle of $\Theta_e$ (Figure 6). In order to further improve the efficiency, a hypotenuse angle $\Theta_e$ was designed at the end of the hollow cavity to reduce the loss caused by the internal bounce of light. Figure 7 shows the illumination map observed for the detector at a $\Theta_e$ value of 45°. Table 3 presents the simulated uniformity and efficiency at different $\Theta_e$ values. In the Table 3 and Figure 7, the efficiency and uniformity were about 86% and 84.4%, respectively, when the $\Theta_e$ was at 45°. The angle of the bevel at the end of the hollow cavity was a single angle, causing the light to concentrate too much in a certain area, resulting in a decrease in uniformity. Therefore, it was necessary to optimize the curvature of the tail end to change to the multi-section curvature. As indicated in the Table 3, the best efficiency was observed at a $\Theta_e$ value of 45°. However, the uniformity decreased to 84.4% at this angle. We could not achieve favorable efficiency and uniformity simultaneously at the same $\Theta_e$ value. Therefore, we proposed the use of VARS on the end wall in the diffuse reflection cavity.

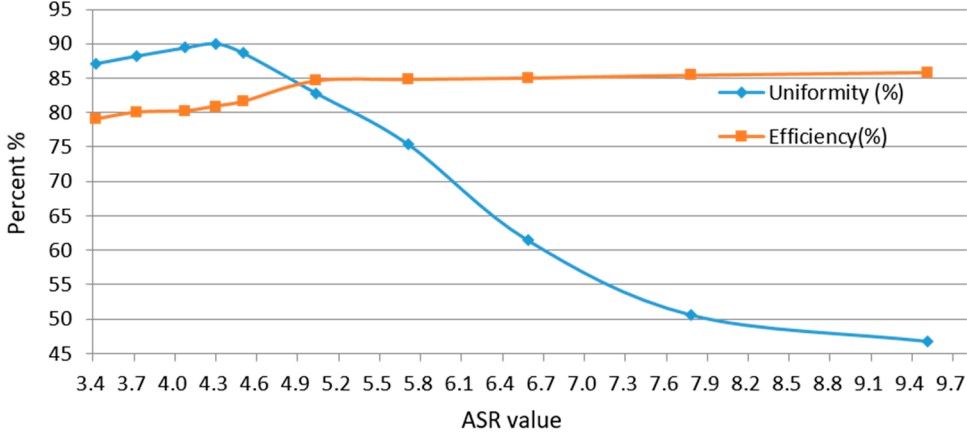

**Figure 4.** Relationship between efficiency and uniformity at various aspect ratio values for hollow light guide.

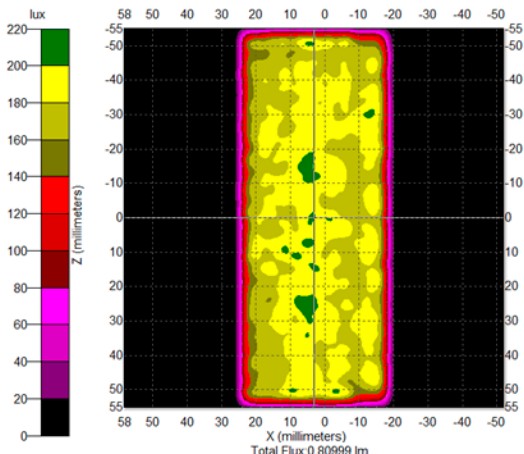

**Figure 5.** Illumination map observed by a detector.

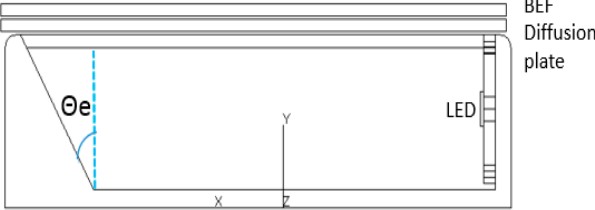

**Figure 6.** Lateral section of the light guide module that show a tilting end wall at an angle of $\Theta_e$.

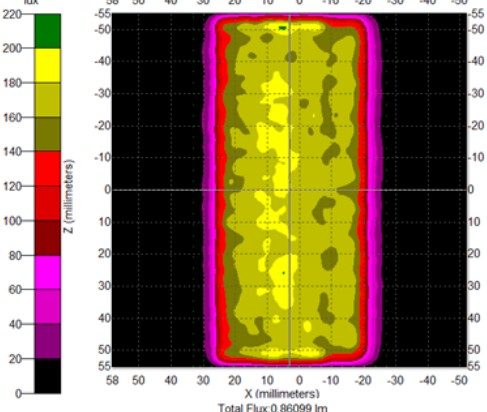

**Figure 7.** Illumination map of a light guide module measured by detector at $\Theta_e = 45°$.

**Table 3.** Simulated efficiency and uniformity at different $\Theta_e$ values.

| Tilting end Wall Angle ($\Theta_e$) | Uniformity (%) | Efficiency (%) |
|---|---|---|
| 20° | 84.6 | 85.3 |
| 30° | 84.5 | 85 |
| 45° | 84.4 | 86 |
| 60° | 82.2 | 85 |

## 2.3. Optimized VARS

Figure 8 illustrates the VARS implemented on the end wall. It shows the VARS structure design, which was used to adjust the uniformity distribution of the light emitted from the hollow cavity. The main idea of this study was that the light shone out in the middle of the hollow body. Figure 9 presents the ray tracing in VARS that is an explanation for the derivation of the VARS theory.

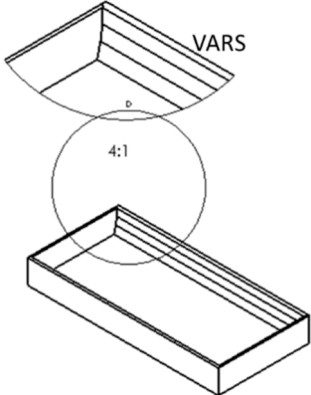

**Figure 8.** Varied angles of the reflection surface (VARS) implemented on the end wall.

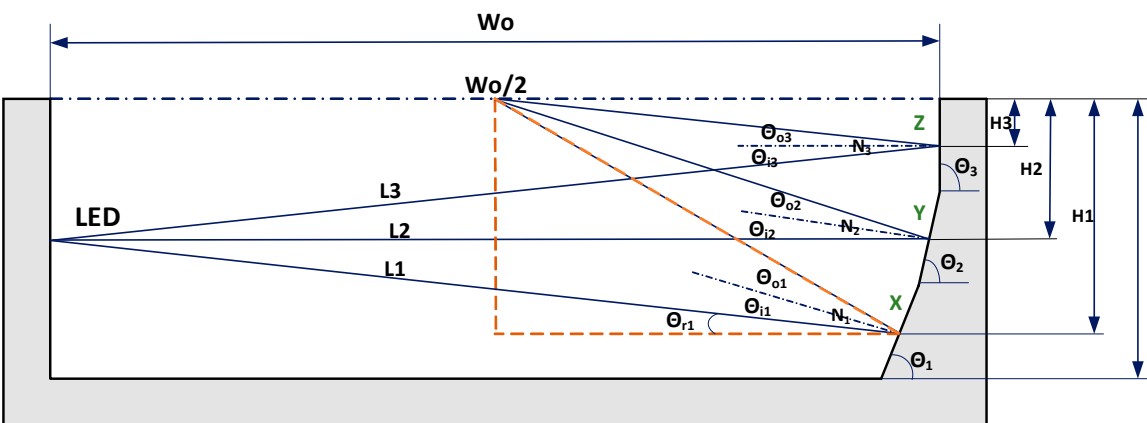

**Figure 9.** Geometry structure of ray tracing with VARS.

To optimize efficiency and uniformity, we let the L1, L2, and L3 rays travel toward the center of the diffusion plate through the reflection engendered by the VARS. In Figure 9, $W_0$ represents the width, and $H_0$ represents the height of the VARS. Moreover, $N_1$–$N_3$ denote the normal lines perpendicular to the X–Z sections, respectively. The tilt angles $\Theta_1$, $H_1$, and $\Theta_{r1}$ are defined in Equations (2)–(4). $\Theta_{i1}$ is the incident angle, $\Theta_{r1}$ is the reference angle of $\Theta_1$, and $\Theta_{o1}$ is the reflection angle corresponding to $N_1$ in the X section.

$$\tan \Theta_{r1} = \frac{H_0 - \frac{H_0}{6}}{\frac{W_0}{2}},$$

(2)

$$H_1 = H_0 - \frac{H_0}{6}, \tag{3}$$

$$\boldsymbol{\Theta_1} = 90 - \boldsymbol{\Theta_{r1}} - \boldsymbol{\Theta_{i1}}. \tag{4}$$

The second tilt angle $\boldsymbol{\Theta_2}$ and $H_2$ are defined in Equations (5) and (6). $\boldsymbol{\Theta_{i2}}$ is the incident angle, and $\boldsymbol{\Theta_{o2}}$ is the reflection angle corresponding to $N_2$ in the Y section.

$$\boldsymbol{\Theta_2} \cong 90 - \boldsymbol{\Theta_1}, \tag{5}$$

$$H_2 = H_0 - \frac{3H_0}{6}. \tag{6}$$

The third tilt angle $\boldsymbol{\Theta_3}$ and $H_3$ are defined in Equations (7) and (8). $\boldsymbol{\Theta_{i3}}$ is the incident angle, and $\boldsymbol{\Theta_{o3}}$ is the reflection angle corresponding to $N_3$ in the Z section.

$$\boldsymbol{\Theta_3} \cong 90 - \boldsymbol{\Theta_I}, \tag{7}$$

$$H_3 = H_0 - \frac{5H_0}{6}. \tag{8}$$

Through Equations (2)–(8), VARS profiles can be drawn, and VARS can be defined by the following polynomial equation:

$$Y = C_0 + \sum B_i * X^i \ i = 0, \ 1, \ 2 \ldots \tag{9}$$

where $C_0$ is a constant, and B1–B6 are the coefficients of the polynomial equation listed in Table 4.

**Table 4.** Coefficients of polynomial equation.

| Symbol | Value |
|---|---|
| Intercept ($C_0$) | $-3.49128 \times 10^{-14}$ |
| B1 | 16.708 |
| B2 | −48.53044 |
| B3 | 56.33018 |
| B4 | −18.82259 |
| B5 | −4.49559 |
| B6 | 2.73919 |

Figure 10 shows the proposed hollow light guide with VARS. It illustrates a 3D schematic diagram of a hollow cavity combined with VARS. Figure 11 illustrates the illumination map observed for the proposed hollow light guide with VARS. The uniformity was determined to be 89%, and the overall efficiency was determined to be 85%. It can be seen from the light trace that the light was concentrated in the middle of the hollow cavity through designing multi-segment VARS. At this time, the uniformity was 89%, and the efficiency of 85% had reached a good result. In order to improve overall uniformity, the design of a beveled reflection surface at the entrance of the hollow cavity could converge the LED light intensity distribution at the module.

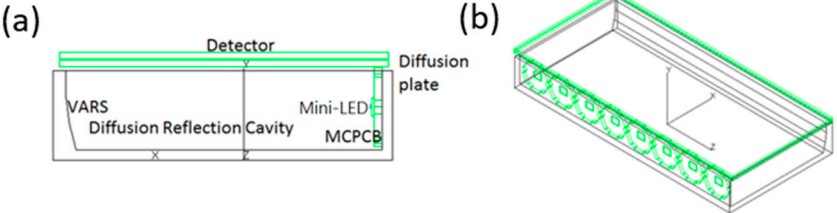

**Figure 10.** Hollow light guide with VARS: (**a**) side view and (**b**) isometric view.

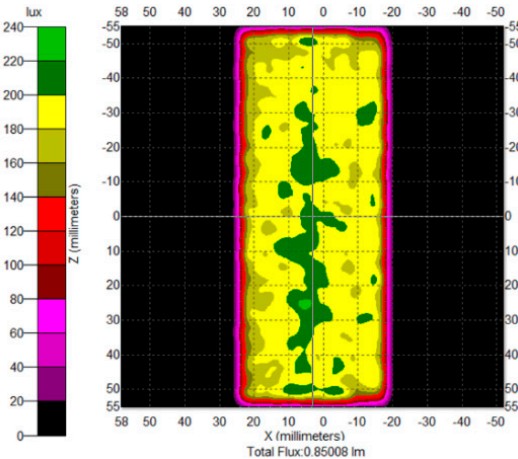

**Figure 11.** Illumination map of a novel hollow light guide with VARS.

## 2.4. Beveled Reflection Surface Design and Optimization

As indicated in Figure 11, the luminance at the center of the detection surface was higher than the overall luminance because the light was directed to the center through the VARS. We optimized $\Theta_4$ to further enhance uniformity. A beveled reflection surface was designed to reduce the vertical axis beam angle of the mini-LEDs to enhance uniformity. Table 5 lists the efficiency and uniformity measured at various angles of reflection bevel edge $\Theta_4$.

**Table 5.** Efficiency and uniformity of calculation results at various bevel angles $\Theta_4$.

| Reflection Bevel Edge ($\Theta_4$) | Uniformity (%) | Efficiency (%) |
|:---:|:---:|:---:|
| 10° | 86.7 | 80.7 |
| 20° | 88.2 | 81.1 |
| 30° | 90.8 | 82.5 |
| 40° | 90.6 | 81.7 |
| 50° | 87.1 | 81.3 |

Figure 12 illustrates the illumination map measured for the beveled reflection surface. After optimization, it can be seen from Table 5 and Figure 12 that the efficiency can be increased to 82.5% and the uniformity can reach 90.8% when the $\Theta_4 = 30°$. Figure 13 shows the intensity distribution curve for the beveled reflection surface in the detector. Figure 13 is the light distribution curve after the simulation, wherein the blue line is a 0° tangent angle, the green line is a 45° tangent angle, the red line is a 90° tangent angle, and the cyan line is a 135° tangent angle light distribution curve, respectively.

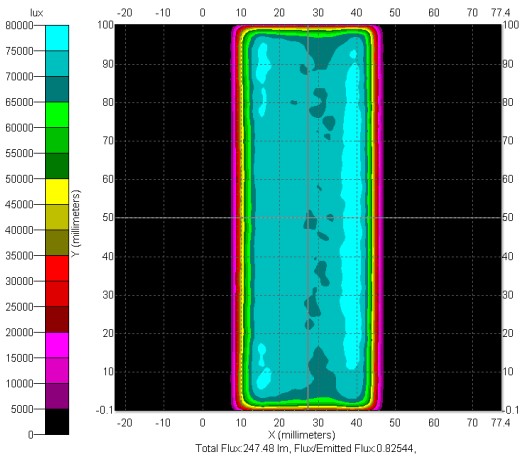

**Figure 12.** Illumination map for beveled reflection surface.

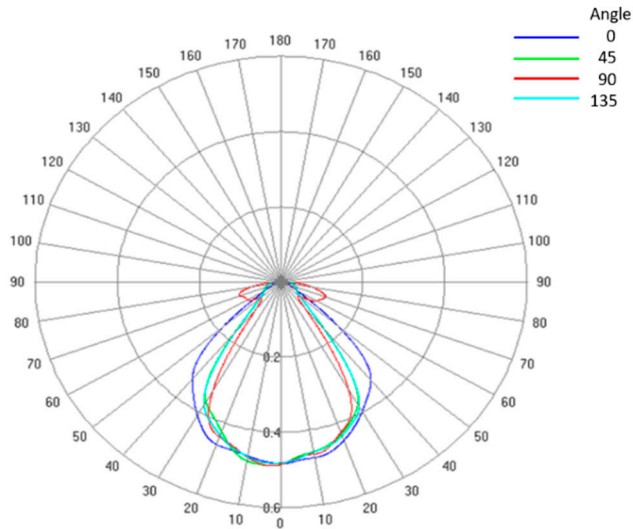

**Figure 13.** Intensity distribution curve for the beveled reflection surface.

## 3. Actual Production and Verification

Figure 14 is the actual finished product which depicts an archetype of the planar mini-LED light source module with VARS and a beveled reflection surface. Figure 15 shows the measured intensity curve obtained using the LEDGON goniometer instrument (Instrument Systems GmbH, Munich, Germany). Approximation and simulation data are presented in Figure 13. Figure 16a presents the illumination map measured when the module was shut down. Figure 16b shows the illumination map when the module was in operation. Nine equidistant points were established to evaluate uniformity, and Figure 17 shows these measurement points obtained using a BM-7 luminance meter (Topcon Corporation, Tokyo, Japan). Uniformity was evaluated by determining (Lmin/Lmax × 100%). Table 6 lists the measurement results. The data presented in Table 6 were measured using an integrating sphere (Isuzu Optical, Hsinchu, Taiwan). The relative luminance (L) measured at point 1 was 92.3%. Table 7 presents the measured optical characteristics of the module. The module efficiency was 83.9%. Table 8 provides a comparison of the simulation and measurement results, revealing close agreement.

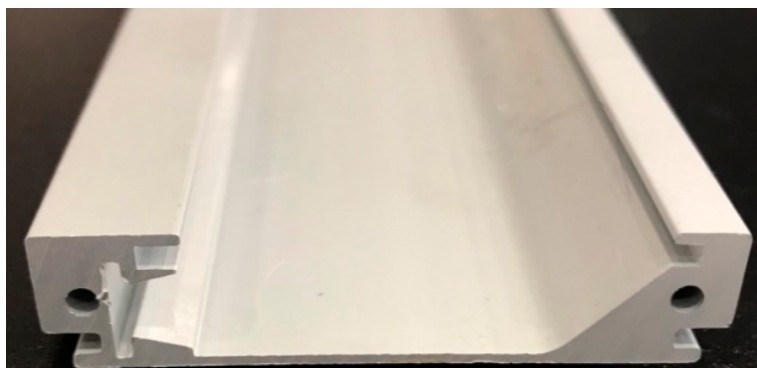

**Figure 14.** Archetype of light source module with VARS and beveled reflection surface.

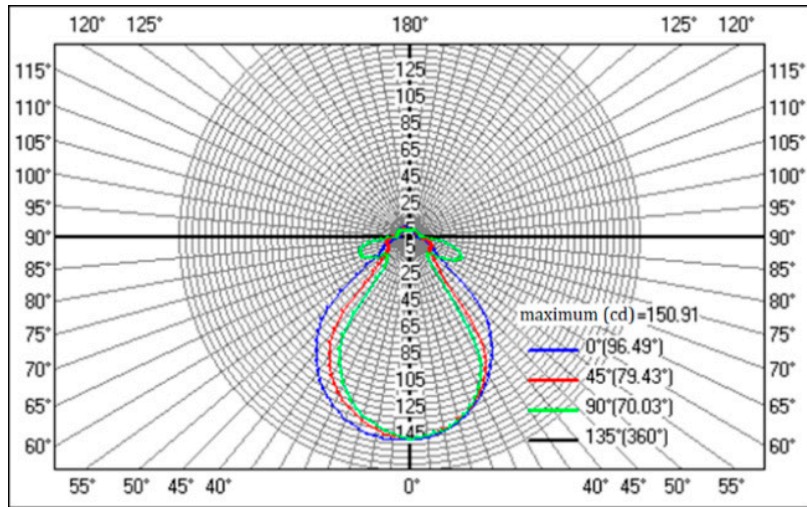

**Figure 15.** Intensity distribution of actual light guide module.

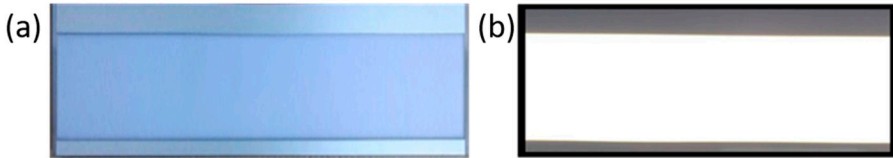

**Figure 16.** Illumination conditions of proposed light source module: (**a**) shut down and (**b**) in operation.

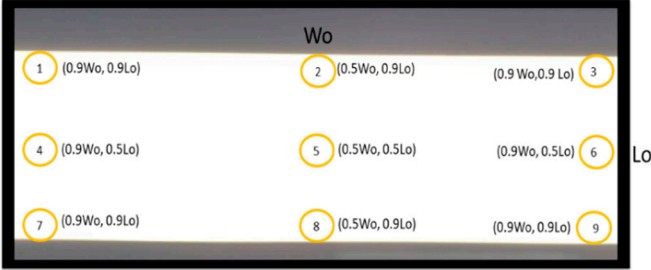

**Figure 17.** Measurement points on light source module for uniformity evaluation.

**Table 6.** Relative luminance measured at nine points.

| Position | Relation Luminance L (nits) |
|----------|------------------------------|
| $P_1$ | 0.925 |
| $P_2$ | 0.923 |
| $P_3$ | 0.927 |
| $P_4$ | 0.984 |
| $P_5$ | 1 |
| $P_6$ | 0.986 |
| $P_7$ | 0.936 |
| $P_8$ | 0.931 |
| $P_9$ | 0.938 |

**Table 7.** Optical characteristics of light source and light source module.

| | CIE x | CIE x | Flux (lm) | Efficiency (%) |
|---|-------|-------|-----------|-----------------|
| Light source | 0.3792 | 0.3821 | 300 | 100 |
| Light source module | 0.3802 | 0.3734 | 251 | 83.9 |

**Table 8.** Comparison of simulation and measurement results.

| Item | Simulation | Measurement |
| --- | --- | --- |
| Uniformity (%) | 90.8 | 92.3 |
| Efficiency (%) | 82.5 | 83.9 |
| Horizontal axis (degree) | 104 | 106 |
| Vertical axis (degree) | 76 | 74 |

## 4. Conclusions

This paper presents a novel planar light source module involving mini-LEDs with asymmetric luminous intensity distribution and a novel optical cavity consisting of a diffuse reflection cavity, beveled reflection surface, and VARS to achieve high efficiency and uniformity. The archetype achieved a uniformity of 92.3% and efficiency of 83.9%. We used varied angles of the reflection surface (VARS) on the end wall in the diffuse reflection cavity instead of traditional solid light guide architecture to reduce the thickness and weight of this light source module. This architecture with an asymmetric luminous intensity distribution and diffuse reflection cavity can improve LED lighting performance. The proposed hollow light guide had a width of 51.5 mm, making it more suitable for linear lighting or slender backlighting than for display stand applications.

**Author Contributions:** Z.T.Y. and H.T.N. were responsible for data and optical measurements. C.H.T. was responsible for actual production and verification. C.L.C., L.-C.C., and H.-C.W. organized and wrote the manuscript.

**Funding:** The authors would like to thank the Ministry of Science and Technology of Taiwan for their financial support under grant number MOST107-2221-E-009-113-MY3 and cross-industry integration of biomedical leap project B10801.

**Conflicts of Interest:** The authors declare no conflicts of interest.

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
