# Peer review of "Hollow Light Guide Module Involving Mini Light-Emitting Diodes for Asymmetric Luminous Planar Illuminators"

_energies, doi:10.3390/en12142755_

Round 1
Reviewer 1 Report
Due to the extensive use of three letter abbreviations it is very hard to read the text of the paper. Please, refrain from this prectice; do not be laisy to write the entire expression. Thos most annoying such abbreviation is the "ALI" that is widely known as "avareged LED intensity", defined in section 4.3 of the CIE 127:2007 document. The term abbreviated by ALI in the paper is very similar, therefore, it is confusing.
Please, also improve the English of your paper. E.g. "LEDs are a point light source" is correctly "LEDs are point light sources". The text contains multiple similar ill-formed sentences. Another example: "The disadvantage of an LGP is bulk." I guess, you wanted to say that "The disadvantage of light guide plates is that they are bulky."
From your description I cannot judge the uniqueness nor the novelty of your solution. Perhaps more comapistosn with other technical solutions would enhance your paper.
Author Response
1.Due to the extensive use of three letter abbreviations it is very hard to read the text of the paper. Please, refrain from this prectice; do not be laisy to write the entire expression. Thos most annoying such abbreviation is the "ALI" that is widely known as "avareged LED intensity", defined in section 4.3 of the CIE 127:2007 document. The term abbreviated by ALI in the paper is very similar, therefore, it is confusing.
Reply:
We already have fixed them as in the revised manuscript. These abbreviations are no longer in this manuscript (except a long term VARS), and we try to express the terms coherently.
In detail:
1. The abbreviated term “ALI” was extended to “asymmetric luminous intensity”
2. The abbreviated term “ASR” was changed into “aspect ratio”. We just kept abbreviation in the diagram in figure 4.
3. The abbreviated term “BEF” was changed into “brightness enhancement film”. We just kept abbreviation in figure 1, figure 3, and figure 6.
2.Please, also improve the English of your paper. E.g. "LEDs are a point light source" is correctly "LEDs are point light sources". The text contains multiple similar ill-formed sentences. Another example: "The disadvantage of an LGP is bulk." I guess, you wanted to say that "The disadvantage of light guide plates is that they are bulky."
Reply:
We rewrote some sentences and paragraphs in which they make confused or are unclear meanings.
In detail:
1. “LEDs are a point light source” changed to “LEDs are point light sources”
2. “An LGP is a solid light guide. The common materials used are polycarbonate (PC) and polymethyl methacrylate (PMMA). The disadvantage of an LGP is bulk.” changed to “A light guide plate is mainly made from solid materials such as polycarbonate (PC) and polymethyl methacrylate (PMMA). Using light guide plate encountered several disadvantages due to its bulky characteristics in size and weight.”
3.From your description I cannot judge the uniqueness nor the novelty of your solution. Perhaps more comapistosn with other technical solutions would enhance your paper.
Reply:
There are some improvements of our design
- Reducing the thickness and weight: We used varied angles of the reflection surface (VARS) on the end wall in the diffuse reflection cavity instead of traditional solid light guide architecture.
- Enhance uniformity and efficiency: This architecture with an asymmetric luminous intensity distribution and diffuse reflection cavity can improve the LED lighting performance.
We added reference:
[18] “Xu, S.; Yang, T.; Miao, H.-H.; Xu, Y.-Z.; Shen, Q.-X.; Guo, T.-L.; Cui, Z.-X.; Chen, E.-G.; Ye, Y. Tilted light coupling structure for the thickness reduction of a liquid crystal display backlight. Appl. Opt. 2019, 58, 2567-2574.” in order to express more recently designs of researcher to enhance the uniformity of diffusion plate.
[25] “Xu, P.; Luo, T.-Z.; Zhang, X.-L.; Su, Z.-J.; Huang, Y.-Y.; Li, X.-C.; Zou, Y. Design and optimization of a partial integrated backlight module. Optics Communications 2018, 427, P 589-595” in order to show more evident in traditional design of light guide.
Reviewer 2 Report
This novel planar light source module involving mini-LEDs are well designed, test and analyzed. But need to provide sufficient background and include all relevant references in recent year, else the novelty is not clear. Besides, it is necessary to quantitatively compare the new design and the current planar light source modules. What is the improvement for the new design?
Author Response
This novel planar light source module involving mini-LEDs are well designed, test and analyzed. But need to provide sufficient background and include all relevant references in recent year, else the novelty is not clear. Besides, it is necessary to quantitatively compare the new design and the current planar light source modules. What is the improvement for the new design?
Reply:
There are some improvements of our design
- Reducing the thickness and weight: We used varied angles of the reflection surface (VARS) on the end wall in the diffuse reflection cavity instead of traditional solid light guide architecture.
- Enhance uniformity and efficiency: This architecture with an asymmetric luminous intensity distribution and diffuse reflection cavity can improve the LED lighting performance.
We added reference
[18] “Xu, S.; Yang, T.; Miao, H.-H.; Xu, Y.-Z.; Shen, Q.-X.; Guo, T.-L.; Cui, Z.-X.; Chen, E.-G.; Ye, Y. Tilted light coupling structure for the thickness reduction of a liquid crystal display backlight. Appl. Opt. 2019, 58, 2567-2574.” in order to express more recently designs of researcher to enhance the uniformity of diffusion plate.
[25] “Xu, P.; Luo, T.-Z.; Zhang, X.-L.; Su, Z.-J.; Huang, Y.-Y.; Li, X.-C.; Zou, Y. Design and optimization of a partial integrated backlight module. Optics Communications 2018, 427, P 589-595” in order to show more evident in traditional design of light guide.
Reviewer 3 Report
Although the work has been carefully designed however there ate several drawbacks can be found in this work including
i-not up to date references
ii-uninterresting introduction
iii-less conclusive conclusion
Author Response
Although the work has been carefully designed however there are several drawbacks can be found in this work including
i-not up to date references
Reply:
We have added some references recently years in order to guarantee up to date information.
In detail:
We added reference
[18] “Xu, S.; Yang, T.; Miao, H.-H.; Xu, Y.-Z.; Shen, Q.-X.; Guo, T.-L.; Cui, Z.-X.; Chen, E.-G.; Ye, Y. Tilted light coupling structure for the thickness reduction of a liquid crystal display backlight. Appl. Opt. 2019, 58, 2567-2574.” in order to express more recently designs of researcher to enhance the uniformity of diffusion plate.
[25] “Xu, P.; Luo, T.-Z.; Zhang, X.-L.; Su, Z.-J.; Huang, Y.-Y.; Li, X.-C.; Zou, Y. Design and optimization of a partial integrated backlight module. Optics Communications 2018, 427, P 589-595” in order to show more evident in traditional design of light guide.
ii-uninterresting introduction
Reply:
We already rewrote the introduction by changing paragraphs.
In detail:
“Light-emitting diodes (LEDs) are a typical light source used in lighting and backlighting applications. LEDs have become the main light source for backlighting liquid crystal displays and lighting devices” changed to “After several decades of Light-emitting diodes (LEDs) emergent, LED lighting technology has undergone several significant developments and has gradually become the typical light source for backlighting liquid crystal displays and lighting devices, for which the revolution has marked a milestone breakthrough in the LED technology and led LEDs to become an irreplaceable light source.”
iii-less conclusive conclusion
Reply:
I appreciate reviewer’s comment. We have rewrote as: This paper presents a novel planar light source module involving mini-LEDs with asymmetric luminous intensitydistribution and a novel optical cavity consisting of a diffuse reflection cavity, beveled reflection surface, and VARS to achieve high efficiency and uniformity. The archetype achieved a uniformity of 92.3% and efficiency of 83.9%. We used varied angles of the reflection surface (VARS) on the end wall in the diffuse reflection cavity instead of traditional solid light guide architecture for reducing the thickness and weight of this light source module. This architecture with an asymmetric luminous intensity distribution and diffuse reflection cavity can improve the LED lighting performance. The proposed hollow light guide has a width of 51.5 mm, making it more suitable for linear lighting or slender backlighting than for display stand applications.
Round 2
Reviewer 1 Report
Pleas run a spell-check, since when you applied automatic "find-and-replace" for the three-letter abbreviations you forgot a space from the end.
Author Response
We appreciate reviewer’s comment. We finish a spell-check for this manuscript.
Reviewer 2 Report
Agree to accept it.
Author Response
We appreciate reviewer’s comment.
Reviewer 3 Report
No
Author Response
We appreciate reviewer’s comment.